# Evaluation of PAGE-B Score for Hepatocellular Carcinoma Development in Chronic Hepatitis B Patients: Reliability, Validity, and Responsiveness

**DOI:** 10.3390/biomedicines12061260

**Published:** 2024-06-05

**Authors:** Evanthia Tourkochristou, Maria Kalafateli, Christos Triantos, Ioanna Aggeletopoulou

**Affiliations:** 1Division of Gastroenterology, Department of Internal Medicine, University Hospital of Patras, 26504 Patras, Greece; evanthiatourkohristou@gmail.com (E.T.); chtriantos@upatras.gr (C.T.); 2Department of Gastroenterology, General Hospital of Patras, 26332 Patras, Greece; mariakalaf@hotmail.com

**Keywords:** chronic hepatitis B, hepatocellular carcinoma, PAGE-B score, HCC surveillance, validity, reliability

## Abstract

Chronic hepatitis B (CHB) constitutes a major global public health issue, affecting millions of individuals. Despite the implementation of robust vaccination programs, the hepatitis B virus (HBV) significantly influences morbidity and mortality rates. CHB emerges as one of the leading causes of hepatocellular carcinoma (HCC), introducing a major challenge in the effective management of CHB patients. Therefore, it is of utmost clinical importance to diligently monitor individuals with CHB who are at high risk of HCC development. While various prognostic scores have been developed for surveillance and screening purposes, their accuracy in predicting HCC risk may be limited, particularly in patients under treatment with nucleos(t)ide analogues. The PAGE-B model, incorporating age, gender, and platelet count, has exhibited remarkable accuracy, validity, and reliability in predicting HCC occurrence among CHB patients receiving HBV treatment. Its predictive performance stands out, whether considered independently or in comparison to alternative HCC risk scoring systems. Furthermore, the introduction of targeted adjustments to the calculation of the PAGE-B score might have the potential to further improve its predictive accuracy. This review aims to evaluate the efficacy of the PAGE-B score as a dependable tool for accurate prediction of the development of HCC in CHB patients. The evidence discussed aims to provide valuable insights for guiding recommendations on HCC surveillance within this specific population.

## 1. Introduction

Chronic hepatitis B (CHB) is a global public health problem that affects more than 250 million people worldwide [1]. Despite the presence of well-established vaccination programs in various countries, HBV continues to have a substantial impact on morbidity and mortality rates [2], especially in African and East Asian countries. CHB can progress to cirrhosis, liver failure, and hepatocellular carcinoma (HCC). It stands as one of the predominant etiologies of HCC, contributing to 33% of HCC-related mortality [3]. The risk of HCC persists even in patients with suppressed viral replication, and HCC can manifest in CHB patients irrespective of pre-existing cirrhosis. Consequently, the presence of HCC in CHB poses a significant complication and a central challenge in the management of these patients [4,5]. The timely diagnosis of HCC enhances the feasibility of curative therapies and ultimately improves patients’ prognosis. Hence, the vigilant surveillance of CHB patients and the identification of high-risk HCC patients holds paramount clinical significance [6]. Several prognostic scores have been developed utilizing clinical and laboratory parameters, offering an avenue to enhance surveillance and screening strategies. Many of these risk scores have focused on untreated patients, mostly based on pretreatment risk factors for HCC and HBV DNA levels [7,8]. These models seem not to accurately predict HCC risk in patients treated with nucleos(t)ide analogues (NUCs), potentially leading to an overestimation of HCC incidence [9]. Optimized models developed for those under antiviral treatment, such as modified REACH-B (mREACH-B) and Platelet-Age-Gender-Hepatitis B (PAGE-B), have been developed, yielding promising results. The PAGE-B model was originally created to predict HCC development in Caucasian CHB patients undergoing HBV treatment over a 5-year period [10]; in subsequent studies, its predictive efficacy was confirmed in both European and Asian populations [9,10]. Due to its reliance on affordable and easily accessible measurements, excluding the need for cirrhosis assessment, the PAGE-B score has evolved into a well-established tool for clinicians, particularly in settings where access to liver biopsy or transient elastography (TE) is limited [11]. Uncertainties persist regarding the optimal application of this score in clinical practice, particularly in terms of when and how to utilize it. The comparative prognostic accuracy of noninvasive scores against liver histology remains unknown and has not been extensively assessed. This review assesses the PAGE-B score’s utility as a reliable tool for accurate prediction of HCC development in CHB patients. By reviewing available studies that have explored the reliability and validity of the PAGE-B score, our aim is to contribute to the update of current recommendations for HCC surveillance in this population (Figure 1).

## 2. Current HCC Recommended Surveillance in CHB Patients

Specific guidelines for HCC surveillance in CHB patients have been established, aiming to accurately identify those at the highest risk for HCC development, who need close monitoring. Although all guidelines agree on the necessity for regular surveillance in the presence of cirrhosis, there are grey areas in the risk stratification of HBV patients without cirrhosis or in cirrhotics receiving anti-viral treatment [12]. There is a widespread agreement among liver disease organizations regarding the adoption of abdominal ultrasound scans (USSs) every 6 months as the preferred method for HCC surveillance. The use of serum alpha-fetoprotein (AFP) remains controversial among experts [12]. Several studies have highlighted the limited sensitivity and specificity of AFP in the accurate diagnosis of HCC [13,14]. Various scoring systems have been proposed for the prediction of HCC risk that are easily applicable in clinical practice, have high predictive value, and can facilitate effective surveillance in high-risk patients with minimal monitoring in those classified as low risk. Several HCC risk scores have been developed for untreated CHB patients, including GAG-HCC (guide with age, sex, HBV DNA, central promoter mutations, and cirrhosis-hepatocellular carcinoma), NGM-HCC (nomogram-hepatocellular carcinoma), and REACH-B (risk estimate for hepatocellular carcinoma in chronic hepatitis B) [8,15,16], as well as for treated CHB patients, such as mREACH-B, PAGE-B (platelets, age, sex, and HBV), mPAGE-B, CAMD (cirrhosis, age, male sex, and diabetes mellitus), and REAL-B (Asia-Pacific Rim real-world efficacy for HBV risk scoring) [10,17,18]. Additionally, models that encompass mixed patient populations with varying treatment status have been developed [CU-HCC (Chinese University-hepatocellular carcinoma), LSM-HCC (liver stiffness measurement-hepatocellular carcinoma), and RWS-HCC (real-world risk score-hepatocellular carcinoma)] [19]. None of these models have received widespread recommendations for use in routine clinical practice according to the established guidelines. Zeng et al. suggest the adoption of contemporary HCC risk scores to enhance HBV patient surveillance, with mREACH-B as the preferred model in patients under antiviral treatment and PAGE-B in cases of non-available liver stiffness measurements [12]. The authors also suggest specific cut-offs for these scores that stratify patients in different risk groups, thus further prioritizing those with urgent HCC surveillance.

## 3. The PAGE-B Model

The PAGE-B model, introduced in 2016 by Papatheodoridis et al., was designed to forecast the development of HCC over a 5-year period in 1815 Caucasian patients undergoing HBV treatment, including entecavir (ETV) or tenofovir disoproxil fumarate (TDF) [10]. For the development of the PAGE-B model, a derivation dataset including patients from eight different centers served as the training dataset. Subsequently, a validation dataset, including patients from the largest center in Milano, Italy, was employed for external validation of the scoring system. All adult patients (≥16 years old) with CHB who were Caucasians and had received treatment with ETV or TDF for a duration of at least 12 months were included in the study [10]. The 5-year cumulative incidence rates of HCC were 5.7% in the derivation dataset and 8.4% in the validation dataset. In the derivation dataset, age, gender, platelet count, and cirrhosis were identified as independent predictors of HCC [10]. The development of the PAGE-B score, based on age, gender, and platelets, yielded a concordance index (c-index) of 0.82, remaining robust at 0.81 after bootstrap validation. The addition of cirrhosis did not significantly improve predictive discrimination (c-index: 0.84). The predictability of the PAGE-B score remained consistent in the validation dataset, with a c-index of 0.82. Stratifying patients by specific PAGE-B cut-offs (<10, 11–17, >17), the 5-year cumulative HCC incidence rates were 0%, 3%, and 17% in the derivation dataset and 0%, 4%, and 16% in the validation dataset, respectively [10].

## 4. Accuracy of the PAGE-B Score in Predicting HCC Risk in CHB Patients

The diagnostic accuracy of the PAGE-B score in HCC prediction was explored in subsequent studies. A retrospective study conducted by Gokcen et al. aimed to assess the accuracy of the PAGE-B score in predicting the risk of HCC among 742 Turkish CHB patients undergoing TDF or ETV therapy for a minimum of 1 year [20]. The mean follow-up for these patients was recorded as 54.7 ± 1.2 months. The accuracy of the PAGE-B score in predicting HCC risk was assessed using a time-dependent area under the receiver operating characteristic (AUROC) curve at all points throughout the study. During the study period, HCC was diagnosed in 26 patients (3.5%). The cumulative HCC incidence at 1, 3, 5, and 10 years was observed as follows: 0%, 0%, 0%, and 0.4% in the PAGE-B low-risk group; 0%, 1.2%, 1.5%, and 2.1% in the PAGE-B moderate-risk group; and 5%, 11.7%, 12.5%, and 15% in the PAGE-B high-risk group, respectively. The AUROC values of the PAGE-B score in predicting HCC development at 1, 3, 5, and 10 years were recorded as 0.977, 0.903, 0.903, and 0.865, respectively [20].

In another longitudinal retrospective cohort study conducted at the Liver Studies Center of the University Hospital of the Federal University of Maranhão, the primary objective was to assess the accuracy of the PAGE-B and REACH-B scores in predicting the risk of HCC development in 978 CHB patients (males: 45.7%; median age: 47 years; cirrhosis: 18.3%). Among them, 34 patients were diagnosed with HCC. The ROC curve for the PAGE-B score yielded a value of 0.78 (0.79 for REACH-B). The PAGE-B cutoff determined to maximize sensitivity was 11 points [21]. Further examination of the ROC curve highlighted an improved AUC with increasing years of follow-up. The authors concluded that patients scoring below 11 points on the PAGE-B scale may warrant longer screening periods compared to conventional practice [21].

Bollerup et al. sought to evaluate the prognostic accuracy of PAGE-B in a nationwide register-based CHB population (mainly non-cirrhotics) in Denmark [22] which is considered heterogenous regarding country of origin and HBV genotyping. Of the 6016 CHB individuals included in the study, 33 patients (10 with cirrhosis at baseline) developed HCC over a median follow-up period of 7.3 years. This translated to a five-year cumulative incidence of 7.1% (95% confidence interval (CI): 2.0–12.3) for those with baseline cirrhosis and 0.2% (95% CI: 0.1–0.4) for those without cirrhosis [22]. PAGE-B measurement was performed in 1529 patients with available platelet count: in this sub-population, the 5-year cumulative HCC incidence was 0%, 0.8%, and 8.7% for PAGE-B scores of <10, 10–17, and >17, respectively. Harrell’s c-statistic reached 0.91. The PAGE-B cutoff, yielding the highest combined specificity and sensitivity for HCC, was 14 (sensitivity: 82.4%, specificity: 89.4%, positive predictive value (PPV): 5.9%, negative predictive value (NPV): 99.8%). Individuals with PAGE-B scores < 14 at baseline exhibited a five-year cumulative HCC incidence of 0.2%, while those with PAGE-B scores ≥ 14 demonstrated a higher incidence at 5.8% [22].

The optimal utilization and timing of PAGE-B score in clinical practice remains unclear. A comprehensive territory-wide cohort study in Hong Kong aimed to explore the utility of PAGE-B in the identification of patients suitable for exclusion from HCC surveillance while under NUC treatment [23]. The study encompassed 32,150 patients with CHB; over a median follow-up period of 3.9 years, 1532 patients (4.8%) developed HCC. The AUROC for HCC prediction at 5 years was 0.77 for PAGE-B. PAGE-B scores stratified 9417 patients (29.3%) as having low risk for HCC, with a 5-year cumulative HCC incidence of 0.6% [23]. Considering the low NPV (99.5%) of this stratification, the authors concluded that patients stratified as low HCC risk under the PAGE-B model can be safely excluded from HCC surveillance [23].

The predictive accuracy of the PAGE-B score has been also assessed in CHB patients co-infected with HIV. Surial et al. focused on evaluating the predictive performance of the PAGE-B score for HCC development over a 15-year period in patients receiving antiretroviral therapy containing tenofovir (*n* = 2963 individuals with HIV/HBV coinfection) [24]. The distribution of PAGE-B score was as follows: <10 in 26.5%, 10–17 in 57.7%, and ≥18 in 15.7% of patients. Over a median follow-up of 9.6 years, HCC occurred in 68 individuals. The regression slope of the prognostic index for developing HCC within 15 years was 0.93, and the pooled c-index was 0.77, both indicating robust model discrimination [24]. A PAGE-B cut-off of <10 demonstrated a remarkable NPV of 99.4% for HCC development within 5 years. All abovementioned findings validated the utility of PAGE-B as a robust prognostic tool for HCC development in HIV/HBV-coinfected cohorts.

Riveiro-Barciela et al. investigated the utility of the PAGE-B score in the real-world setting, including 611 Caucasian CHB patients (32% cirrhosis) receiving tenofovir or entecavir for more than 4 years [25]. Fourteen individuals (2.29%) developed HCC during the follow-up period (nine diagnosed within the initial 5 years of treatment), all of whom had a baseline PAGE-B score ≥ 10. PAGE-B demonstrated a c-index of 0.732 for HCC prediction at 5 years [25] (Table 1).

## 5. Modifications of PAGE-B Score and Predictive Utility

Some studies have assessed the clinical efficacy of specific modifications of the PAGE-B score in the prediction of HCC risk in CHB patients.

Kim et al. [17] introduced a modification to the PAGE-B model by incorporating the level of serum albumin, resulting in the creation of the mPAGE-B score. This score was originally developed from 2001 Asian CHB patients receiving tenofovir or entecavir and was internally and externally validated (*n* = 1000) [17]. It showed a high discriminative ability, with an AUROC of 0.82 (95% CI: 0.76–0.88) for the validation set, scoring better than other predictive models, including PAGE-B, which had an AUROC of 0.72 (95% CI: 0.65–0.78) in this study (*p* < 0.01).

Subsequently, the effectiveness of PAGE-B and mPAGE-B scores were assessed among 224 CHB Brazilian patients over a median follow-up period of 9 years. The cumulative incidence of HCC at 3, 5, and 7 years was 0.993%, 2.70%, and 5.25%, respectively [26]. HCC development was independently associated with older age, male gender, and the presence of cirrhosis at the time of HBV diagnosis. The AUROCs of PAGE-B and mPAGE-B were 0.7906 and 0.7904, respectively, with no statistically significant differences between them [26]. The high NPV of both PAGE-B (99.14%) and mPAGE-B (98.58%) emphasized the effectiveness of both scoring systems in correctly identifying individuals at low risk for HCC in this context.

In a recent retrospective multicenter cohort study, the authors investigated if the addition of liver stiffness (LS) measurements with transient elastography in the PAGE-B and mPAGE-B scores could increase their discriminative ability in predicting HCC, taking into consideration the well-accepted role of liver fibrotic burden on HCC development in the setting of CHB under antiviral treatment [27]. The study encompassed a cohort of 2184 Korean CHB patients who initiated antiviral treatment (the derivation dataset comprised 1211 patients, while the validation dataset included 973 patients) [27]. Upon identifying independent associations between older age, male sex, lower platelet count, and higher LS values with an increased risk of HCC development, the researchers created a modified PAGE^LS^-B model with a maximum score of 34 [27], which incorporated all the above variables. The time-dependent integrated AUC (iAUC) value for the modified PAGE^LS^-B model reached 0.760 in the derivation dataset, surpassing the AUC values for both PAGE-B (0.714) and mPAGE-B models (0.716); however, in the validation set the iAUC of PAGE-B was higher than that of both mPAGE-B and PAGE^LS^-B [27]. Moreover, the PAGE^LS^-B score could efficiently discriminate patients as having low (score < 12), intermediate (12–24), or high risk (>24) of HCC development. The authors concluded that this new score model could be used as an HCC surveillance tool in CHB patients receiving antiviral treatment, with at least similar discriminative ability to the PAGE-B and mPAGE-B models.

In another nationwide multicenter retrospective study (*n* = 1183), Kaneko et al. aimed to develop a new prognostic model for HCC development based on PAGE-B score and HBV-DNA levels in NUC-treated CHB patients, taking into consideration the predictive role of detectable viral load, even in low levels, in HCC risk [28]. In the multivariate analysis, advanced age, male sex, decreased platelet count, and detectable HBV DNA levels during antiviral treatment were independent predictors of HCC development [28]. The authors then investigated if the detectable HBV DNA could better stratify the patients that had an intermediate score when assessed with the PAGE-B model [28]. Following the observation that patients in both the intermediate and the high PAGE-B risk group with detectable viral load had greater risk for HCC than patients with undetectable HBV DNA, the authors created a new model—the PAGE-B-DNA—which further stratified patients into five risk categories.

In a recent study [29], Chun et al. aimed to develop a new HCC risk model in 3585 Hbe-Ag positive, non-cirrhotic patients who started antiviral treatment at the phase transition from chronic infection to HbeAg (+) chronic hepatitis B, based on baseline HBV DNA levels. In the training cohort (*n* = 2367), independent associations with HCC development were found for age, male sex, platelets, diabetes, and moderate viral load (5.00–7.99 log10 IU/mL) [29]. A new model was developed, named PAGED-B, incorporating these five variables. The new model demonstrated high performance in both training and validation cohorts (time-dependent AUROC: 0.81 and 0.85, respectively) for the prediction of HCC development in 5 years [29]. It remains to be validated in other cohorts from different study groups (Table 2, Figure 2).

## 6. Validating and Applying the PAGE-B Score in Comparison to Other Risk Scores for HCC Surveillance in CHB Patients

The efficacy of HCC risk scores has been studied in Japanese CHB patients receiving NUCs [30]. In a study by Kirino et al., 443 CHB patients with no prior HCC history, treated with entecavir, tenofovir alafenamide, or tenofovir disoproxil fumarate, were recruited. PAGE-B, mPAGE-B, and REACH-B scores were calculated. Over a mean follow-up duration of 5.1 years, 33 patients (7.4%) developed HCC [30]. The AUROC values for PAGE-B at 3 and 7 years (AUROC: 0.786 and 0.744, respectively) consistently surpassed those of REACH-B (AUROC: 0.658 and 0.543) and mPAGE-B models (AUROC: 0.772 and 0.731), implying a superior effectiveness of the PAGE-B model compared to other models among Japanese CHB-treated patients [30].

In another study of 1330 patients that received lamivudine, ETV, or TDF as their initial antiviral regimen, the mPAGE-B score was compared with other HCC risk scores regarding predictive performance [31]. The HCC development rate was 9.6% (*n* = 128) during the follow-up period. The c-indexes of mPAGE-B, GAG-HCC, PAGE-B, REACH-B, and CU-HCC were 0.769, 0.751, 0.744, 0.686, and 0.618, respectively [31]. Although the mPAGE-B score demonstrated the highest c-index, there was no statistical difference when mPAGE-B performance was compared with that of PAGE-B and GAG-HCC models (significant difference was observed when mPAGE-B was compared with REACH-B and CU-HCC) [31]. Thus, this study implies that the addition of albumin levels to the PAGE-B score marginally improves the predictive performance of PAGE-B, and the authors suggested either a statistically improved weight for the variable “albumin” in the mPAGE-B formula or the modification of the model with the incorporation of non-invasive markers of fibrosis. 

Another study conducted by Brouwel et al. sought to evaluate the prognostic efficacy of the PAGE-B score in CHB patients (*n* = 557) in comparison with other conventional scores [32]. They specifically aimed to determine whether incorporating liver histological characteristics could enhance the prognostic accuracy of these HCC risk scores [32] regarding the development of any clinical event, i.e., HCC development, liver failure, transplantation, and mortality. The PAGE-B score exhibited robust predictive performance for any clinical event (c-index: 0.86), HCC development (c-index: 0.91), and transplant-free survival (c-index: 0.83). Compared to the other HCC risk scores that were evaluated (namely FIB-4, REACH-B, log APRI, GAG-HCC, and CU-HCC), PAGE-B showed the highest performance for all outcomes. The discriminative performance of PAGE-B persisted even after stratification by ethnicity, initiation of antiviral therapy post-biopsy, and the presence of advanced fibrosis [32]. When Ishak fibrosis stage was incorporated into the PAGE-B score, only a slight improvement of the c-index was observed (0.87) for any clinical event.

In a French prospective study of 317 CHB patients with biopsy-confirmed Child-Pugh A cirrhosis, PAGE-B and REACH-B were validated [33]. Both models had 100% negative predictive values. PAGE-B had the highest discriminative ability at 1 year (AUROC: 0.808 vs. 0.629, *p* < 0.001; Harrell’s c-index: 0.768 vs. 0.676, *p* = 0.034), but no difference between the two scores was depicted at 3 years [33]. 

Alterations in HCC risk scores and their predictability during NUC treatment were assessed in a cohort comprising 432 patients with no prior history of HCC [34]. PAGE-B, mPAGE-B, and REACH-B scores were computed at the initiation of NUCs, as well as at 1 and 2 years post-treatment [34]. The cumulative incidence of HCC development in individuals classified as high risk by the PAGE-B score at the initiation of NUCs, as well as at 1 and 2 years post-NUC treatment, was significantly higher than in those stratified as intermediate and low risk [34]. However, for patients identified as high risk by mPAGE-B and REACH-B at 2 years post-NUC administration, the HCC incidence was comparable to that observed in the intermediate- and low-risk groups [34]. Furthermore, the AUROCs for HCC development with PAGE-B at the initiation of NUCs, and at 1 and 2 years post-administration, were 0.773, 0.803, and 0.737, respectively [34]. When compared with the AUROCs of mPAGE-B and REACH-B at each time point, PAGE-B consistently performed better [34]. This study highlights the efficient durable application of PAGE-B score at initiation and 1 and 2 years of NUC treatment [34].

Mao et al. assessed the efficacy of non-invasive models, namely ALBI, CAMD, PAGE-B, mPAGE-B, and aMAP, in predicting HCC development among 229 patients with HBV-related liver cirrhosis (both compensated and decompensated) under long-term NUC treatment [35]. The AUROC values for ALBI, aMAP, CAMD, PAGE-B, and mPAGE-B scores were 0.512, 0.667, 0.638, 0.663, and 0.679, respectively [35]. No significant AUROC differences were observed between CAMD, aMAP, PAGE-B, and mPAGE-B, and the authors concluded that the assessed HCC risk scores underestimate HCC risk in decompensated cirrhosis and new models should be derived in this setting.

While the majority of data highlight the superiority of the PAGE-B score over conventional scores, there is also evidence that other models overscore PAGE-B in certain settings. Chon et al. sought to assess and compare the HCC predictive accuracy of three risk prediction models—PAGE-B, SAGE-B, and CAGE-B—following 5 years of ETV therapy among 1335 CHB patients [36]. SAGE-B and CAGE-B were developed recently by Papatheodoridis et al. [37] to predict HCC development after 5 years of antiviral treatment and include 5-year post-treatment variables, in contrast to the PAGE-B score, which contains variables at baseline before treatment initiation. All three scores demonstrated independent associations with HCC development after 5 years of treatment in the multivariate analysis [36]. However, the AUROCs of the SAGE-B and CAGE-B models were significantly higher compared to the PAGE-B model in predicting HCC development after 5 years of entecavir treatment, suggesting that PAGE-B might not be efficiently predictive at this time point [36].

The superior performance of CAGE-B and SAGE-B scores compared to PAGE-B was also depicted in another study [38] of 1763 CHB patients receiving long-term antiviral treatment. The CAGE-B score exhibited the most robust performance, achieving the highest iAUC of 0.820. It was closely followed by the SAGE-B score, with an iAUC of 0.804, and then by mREACH-B (0.800), CAMD (0.786), mPAGE-B (0.748), and PAGE-B (0.721) scores in descending order of predictive accuracy [38]. When statistically compared, the CAGE-B score showed higher performance than SAGE-B, CAMD, PAGE-B, and mPAGE-B scores but was similar to mREACH-B. The SAGE-B score exhibited significantly better performance than mPAGE-B and PAGE-B but was similar to CAMD and mREACH-B scores.

The CAMD scoring system, which was developed in Taiwan to predict HCC development based on the presence of cirrhosis, age, male sex, and diabetes status [18], was compared with PAGE-B and mPAGE-B scores in a Korean study of 3277 CHB patients treated with ETV or TDF [39]. The integrated AUCs of CAMD, PAGE-B, and mPAGE-B for HCC development were 0.790 (0.765–0.813), 0.769 (0.746–0.792), and 0.760 (0.736–0.783), respectively. When comparison was performed among them, the CAMD score demonstrated higher predictive ability compared to PAGE-B but not compared to the mPAGE-B score.

In a multicenter investigation conducted by Lee et al., a total of 3026 previously treatment naive CHB patients who received antiviral treatment for more than 18 months were systematically enrolled [40,41]. The study aimed to assess and validate the efficacy of the recently proposed FSAC prediction model [41], which uses on-treatment modifications of non-invasive fibrosis markers such as APRI and FIB-4 following 12 months of antiviral therapy [40] together with age, gender, and presence of cirrhosis. Harrell’s c-index of the FSAC score (0.770) demonstrated superior discriminative ability, surpassing the corresponding values of PAGE-B (0.725), mPAGE-B (0.738), mREACH-B (0.737), LSM-HCC (0.734), and CAMD (0.742) [40] (Table 3, Figure 3 and Figure 4).

## 7. Cost Evaluation of PAGE-B Score

The assessment of the cost implications associated with the PAGE-B score has also been investigated. In a comprehensive observational single-center study conducted by Sprinzl et al., the potential cost reduction of implementing PAGE-B-tailored HCC ultrasound screening protocols for enhanced cost-effectiveness was explored [42]. The study encompassed 607 CHB patients (227 PAGE-B eligible); 15.8% of them were classified as low HCC risk according to PAGE-B. HCC screening through sonography, with a median examination time of 12.4 min, incurred total costs of EUR 22.82 per examination. Supplementary expenses stemming from patients’ lost earnings or productivity were in the range of EUR 15.6 to 17.5 and EUR 26.7 per examination, respectively [42]. After implementing PAGE-B tailored HCC screening, the annual overall costs for CHB patients could be lowered by 15.51%, corresponding to a 1.91% reduction in the total expenditures of the sonography. The automated computation of the PAGE-B score, along with its readily available components, positions it as a nearly cost-neutral tool for mitigating sonography expenses. Implementing the PAGE-B score-based screening also serves to safeguard limited personnel resources.

## 8. Conclusions

The PAGE-B score, utilizing basic baseline characteristics such as age, gender, and platelet count, has demonstrated significant reliability in predicting HCC in patients with CHB, offering a cost-effective alternative to conventional approaches. The predictive accuracy of the PAGE-B score, whether expressed as the c-index or the AUROC curve, has consistently demonstrated values surpassing 0.7 in numerous studies and seems to perform better than other HCC risk models. The incorporation into PAGE-B of other variables, such as serum albumin, liver stiffness, and HBV DNA levels, seems to only add a modest enhancement to the predictive ability of the model. Nevertheless, it is important to acknowledge the accumulating evidence suggesting that newly developed HCC risk scores may exhibit better predictive accuracy compared to PAGE-B in certain settings, especially when assessing HCC risk after 5 years of antiviral treatment. Despite the robust screening potential of the PAGE-B score in clinical practice, further studies are warranted to assess whether its predictive efficacy for HCC risk differs across diverse patient settings. It is particularly noteworthy that the effectiveness of the PAGE-B score in predicting HCC has not been thoroughly evaluated in patients of African origin, a population in which CHB is a significant cause of morbidity and mortality. A recent study by Patmore et al. [43] highlighted the association between HCC incidence and varying levels of the PAGE-B score in hepatitis B patients of African descent. Specifically, the risk of HCC was negligible among individuals without advanced fibrosis and a low baseline (m)PAGE-B score.

Therefore, it is imperative that future studies focus on evaluating the applicability of the PAGE-B score and its modifications by incorporating larger sample sizes and a broader range of characteristics in high-risk patients of African descent.

## Figures and Tables

**Figure 1 biomedicines-12-01260-f001:**
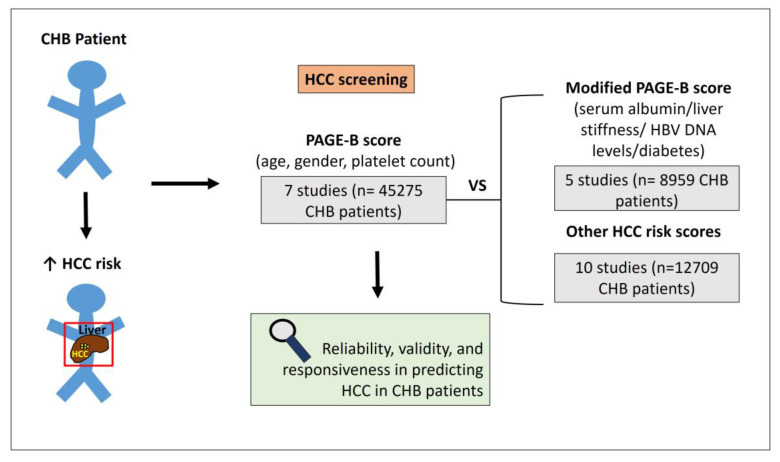
Summary of the pivotal aspects examined within the study.

**Figure 2 biomedicines-12-01260-f002:**
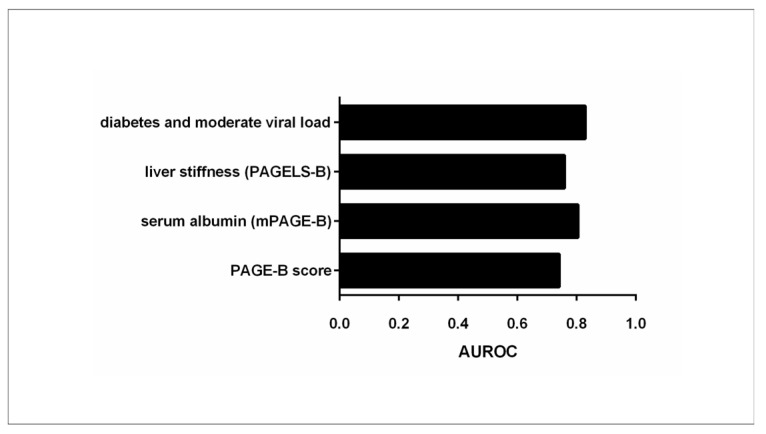
Accuracy of the PAGE-B score in predicting HCC (expressed as AUROC values) compared to modified PAGE-B scores.

**Figure 3 biomedicines-12-01260-f003:**
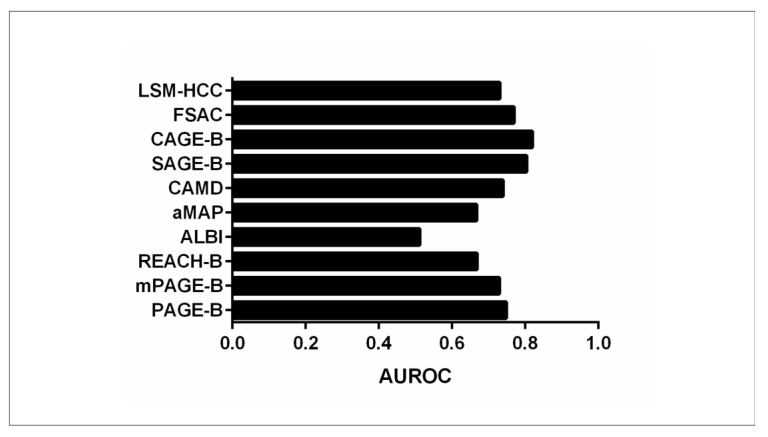
Accuracy of PAGE-B scores in predicting HCC (expressed as AUROC values) compared to other HCC risk scores.

**Figure 4 biomedicines-12-01260-f004:**
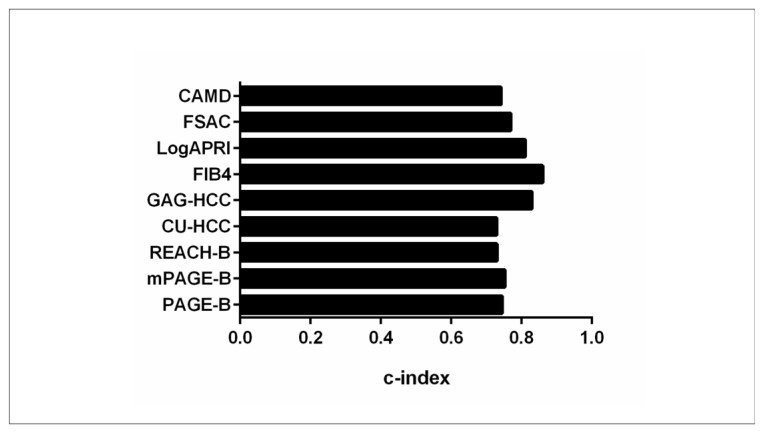
Accuracy of PAGE-B scores in predicting HCC (expressed as c-index) compared to other HCC risk scores.

**Table 1 biomedicines-12-01260-t001:** Accuracy of PAGE-B score in predicting HCC risk.

Reference	CHB Patients (Number, Treatment, Follow-Up Period)	PAGE-B Risk Group Stratification	Accuracy in HCC Prediction
Papatheodoridis et al., 2016 [10]	*n* = 1815, entecavir/tenofovir for ≥12 months, HCC incidence over a 5-year period	<10 (HCC rate: 0%), 11–17 (HCC rate: 4%), >17 (HCC rate: 16%)	C-index = 0.82 in predicting HCC development over a 5-year period PAGE-B risk score ≥ 10 (100% sensitivity and NPV)
Riveiro-Barciela et al., 2017 [25]	*n* = 611, ETV/TDF, mean follow-up was 55 (entecavir) and 49 (tenofovir) months	<10 (low), 10–17 (medium), >17 (high)	PAGE-B score ≥ 10 (sensitivity 100%, specificity 41.2%, NPV 100%, PPV 9.8%)
Yip et al., 2020 [23]	*n* = 32,150, nucleos(t)ide analogue therapy, median follow-up period of 3.9 years	≤9 (low), 10–17 (intermediate), ≥18 (high)	AUROC = 0.77 (5-year HCC prediction) NPV 99.5%
Gokcen et al., 2022 [20]	*n* = 742, tenofovir disoproxil fumarate or entecavir for ≥1 year, mean follow-up time (54.7 ± 1.2 months)	≤9 (low), 10–17 (moderate), and ≥18 (high)	HCC incidences at 1, 3, 5, and 10 years (0%, 0%, 0%, and 0.4% low PAGE-B, 0%, 1.2%, 1.5%, and 2.1% moderate PAGE-B, 5%, 11.7%, 12.5%, and 15% high PAGE-B) AUROC values at 1, 3, 5, and 10 years (0.977, 0.903, 0.903, and 0.865)
Costa et al., 2022 [21]	*n* = 978, entecavir/tenofovir treatment (*n* = 386), medical follow-up for at least 5 years	<11 (HCC patients: 1 out of 274), ≥11 (HCC patients: 7 out of 197)	AUROC curve = 0.788 PAGE-B ≥ 11 (sensitivity 0.875, specificity 0.582, NPV 0.996)
Bollerup et al., 2022 [22]	*n* = 6016, median follow-up period of 7.3 years	<10 (HCC rate: 0%), 10–17 (HCC rate: 0.8%), >17 (HCC rate: 8.7%)	Harrell’s c-statistic = 0.91 PAGE-B ≥ 14 (sensitivity 82.4%, specificity 89.4%, NPV 99.8%)
Surial et al., 2023 [24]	*n* = 2963, CHB/HIV coinfection, antiretroviral therapy (tenofovir), over a 15-year follow-up period	<10 in 26.5% of CHB/HIV patients, 10–17 in 57.7%, and ≥18 in 15.7% of patients	prognostic index for developing HCC within 15 years = 0.93 c-index = 0.77 PAGE-B < 10 (NPV 99.4%)

**Table 2 biomedicines-12-01260-t002:** Modified PAGE-B scores.

Reference	CHB Patients (Number, Treatment, Follow-Up Period)	Adjustments to PAGE-B Score	Accuracy in HCC Prediction
Kim et al., 2018 [17]	*n* = 3001, tenofovir/entecavir, median follow-up period 4.1 years	serum albumin (mPAGE-B)	PAGE-B AUROC 0.72 mPAGE-B AUROC 0.82
da Silva et al., 2022 [26]	*n* = 224, interferon/nucleos(t)ide analogues, median follow-up period of 9 years	serum albumin (mPAGE-B)	PAGE-B > 14 (AUROC 0.7906, Sensitivity 93.33%, NPV 99.14%) mPAGE-B > 10.5 (AUROC 0.7904, Sensitivity 86.67%, NPV 98.58%)
Chon et al., 2021 [27]	*n* = 2184, TDF or ETV, median 43.2 month follow-up duration	liver stiffness (PAGELS-B)	PAGELS-B ≥ 24 (AUC 0.760) PAGE-B (AUC 0.714) mPAGE-B (AUC 0.716)
Kaneko et al., 2020 [28]	*n* = 1183, nucleos(t)ide analogues, median follow-up of 4.9 years	HBV DNA levels (PAGE-B-DNA)	PAGE-B-DNA 10–17 and >18 → ↑ HCC incidence at 3, 5, 7, and 10 years vs. undetectable HBV DNA
Chun et al., 2024 [29]	*n* = 2367, TDF or ETV, median follow-up of 5.4 years	age, male sex, platelets, diabetes, and moderate viral load (5.00–7.99 log10 IU/mL)	time-dependent AUROC: 0.81 and 0.85 in training and validation cohorts

**Table 3 biomedicines-12-01260-t003:** Comparative analysis of PAGE-B score and other HCC risk scores.

Reference	CHB Patients (Number, Treatment, Follow-Up Period)	PAGE-B Score	Other HCC Risk Scores
Kirino et al., 2020 [30]	*n* = 443, ETV, TAF or TDF, mean follow-up duration of 5.1 years	AUROC values at 3 and 7 years (0.786 and 0.744)	AUROC values at 3 and 7 years: REACH-B model (0.658 and 0.543) mPAGE-B (0.772 and 0.731)
Lee et al., 2019 [31]	*n* = 1330, lamivudine, ETV or TDF, follow-up of median 62.0 months	Harrell’s c-index (0.744), NPV 100%	Harrell’s c-index and NPV %: mPAGE-B (0.769 and 100) REACH-B (0.686 and 97.1) CU-HCC (0.618 and 95.8) GAG-HCC (0.751 and 98.5)
Brouwer et al., 2017 [32]	*n* = 557, IFN/nucleos(t)ide analogues, follow-up of mean duration of 10.1 years	c-statistic for HCC development PAGE-B (0.91), PAGE-B + Ishak (0.87)	c-statistic for HCC development: REACH-B (0.83) FIB-4 (0.86) Log APRI (0.81) GAG-HCC (0.91) CU-HCC (0.84)
Brichler et al., 2019 [33]	*n* = 317, IFN/nucleos(t)ide analogues/ETV/TDF, median follow-up of 65.2 months	HCC prediction at 1 year: AUROC 0.808, c-index 0.768	HCC prediction at 1 year: REACH-B AUROC 0.629 c-index 0.676
Kirino et al., 2021 [34]	*n* = 432, nucleos(t)ide analogues, median follow-up of 5.1 years	AUROC for HCC prediction 1 and 2 years after HBV treatment: 0.803 and 0.737	AUROC for HCC development 1 and 2 years after HBV treatment: REACH-B (0.725 and 0.653) mPAGE-B (0.734 and 0.678)
Mao et al., 2023 [35]	*n* = 229, nucleos(t)ide analogues, median follow-up of 37 months	AUROC 0.663	AUROC: ALBI (0.512), aMAP (0.667) CAMD (0.638) mPAGE-B (0.679)
Chon et al., 2023 [36]	*n* = 1335, ETV, follow-up period > 5 years	AUROC for HCC prediction after 5 years of treatment: (95% CI 0.696–0.745)	AUROC for HCC prediction after 5 years of treatment: SAGE-B (95% CI 0.772–0.844) CAGE-B (95% CI 0.785–0.838)
Kim et al., 2020 [39]	*n* = 3277, ETV/TDF, median follow-up period of 58.2 months	iAUC 0.760	iAUC: CAMD (0.790) mPAGE-B (0.769)
Ji et al., 2021 [38]	*n* = 1763, nucleos(t)ide analogues, median follow-up period of 72 months	iAUC 0.721	iAUC: CAGE-B (0.820) SAGE-B (0.804) CAMD (0.786) mREACH-B (0.800) mPAGE-B (0.748)
Lee et al., 2022 [40]	*n* = 3026, ETV/TDF/TAF, median follow-up period of 64 months	Harrell’s c-Index 0.725, iAUC 0.718	Harrell’s c-Index and iAUC: FSAC (0.770 and 0.769) mPAGE-B (0.738 and 0.722) mREACH-B (0.737 and 0.724) LSM-HCC (0.734 and 0.731) CAMD (0.742 and 0.743)

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
