# Peer review of "Evaluation of PAGE-B Score for Hepatocellular Carcinoma Development in Chronic Hepatitis B Patients: Reliability, Validity, and Responsiveness"

_biomedicines, 2024, doi:10.3390/biomedicines12061260_

Round 1

Reviewer 1 Report

Comments and Suggestions for Authors

Chronic hepatitis B (CHB) is one of the leading causes of hepatocellular carcinoma (HCC), which poses a major challenge to the effective management of patients with CHB. 

The authors propose the PAGE-B model, including age, sex, and platelet count, showing remarkable accuracy, validity, and repeatability in predicting the occurrence of HCC among CHB patients receiving HBV treatment. 

The proposed manuscript is innovative and addresses a new aspect regarding the potential of the PAGE-B method to further improve the accuracy of predicting the development of HCC in CHB patients. The article is formatted according to the rules of the journal and presented in good English. The title is too long, and too many abbreviations are used, making it difficult to read. The provision of a graphical abstract, as well as the relevant schemes/graphs in items 4, 5, 6 will facilitate and promote readability.

Comments on the Quality of English Language

-

Author Response

Responses to the comments made by Reviewer 1

Chronic hepatitis B (CHB) is one of the leading causes of hepatocellular carcinoma (HCC), which poses a major challenge to the effective management of patients with CHB.

The authors propose the PAGE-B model, including age, sex, and platelet count, showing remarkable accuracy, validity, and repeatability in predicting the occurrence of HCC among CHB patients receiving HBV treatment.

The proposed manuscript is innovative and addresses a new aspect regarding the potential of the PAGE-B method to further improve the accuracy of predicting the development of HCC in CHB patients. The article is formatted according to the rules of the journal and presented in good English. The title is too long, and too many abbreviations are used, making it difficult to read. The provision of a graphical abstract, as well as the relevant schemes/graphs in items 4, 5, 6 will facilitate and promote readability.

We appreciate the reviewer’s insightful comments. The title was specifically chosen to reflect the primary aim of our study: to evaluate the effectiveness of the PAGE-B score in predicting the development of hepatocellular carcinoma (HCC). Our focus was on assessing the score's reliability, validity, and responsiveness in clinical practice.

The abbreviations used in the manuscript mainly refer to other scores compared to the PAGE-B score, as well as indicators of reliability and validity, and different types of patient treatments. These elements are essential for the comprehensive evaluation presented in our study. All abbreviations are clearly explained in parentheses within the text.

Tables 1, 2, and 3, along with Figures 2, 3, and 4 (which summarize the data from the tables), have been included in sections 4, 5, and 6 of the manuscript (pages 5-8,11-14) These sections highlight the key findings from our analyses, emphasizing the effectiveness, reliability, and validity of the PAGE-B score in predicting HCC, both independently and in comparison with other risk scores. To further clarify the main analysis of our study, we have included a Figure 1 in the Introduction section (page 2).

Reviewer 2 Report

Comments and Suggestions for Authors

The authors present a review to evaluate the effectiveness of the PAGE-B score as a reliable tool for the accurate prediction of HCC development in patients with CHB. The analyzed evidence aims to provide valuable information to guide recommendations on HCC surveillance in this specific population.

The article is well-written, but there are certain modifications that could enhance comprehension.

The part of the references that contains helpful information should be removed; only the references themselves are necessary.

The sections created within the article are not very clear. It should start with an introduction and then continue with the review. An outline could also be helpful to better explain the points to be analyzed in the study.

Section 6 is quite confusing to understand. It should be rewritten for better reader comprehension. An outline or table would help for clearer follow-up.

Author Response

Responses to the comments made by Reviewer 2

The authors present a review to evaluate the effectiveness of the PAGE-B score as a reliable tool for the accurate prediction of HCC development in patients with CHB. The analyzed evidence aims to provide valuable information to guide recommendations on HCC surveillance in this specific population.

The article is well-written, but there are certain modifications that could enhance comprehension.

The part of the references that contains helpful information should be removed; only the references themselves are necessary.

The sections created within the article are not very clear. It should start with an introduction and then continue with the review. An outline could also be helpful to better explain the points to be analyzed in the study.

Section 6 is quite confusing to understand. It should be rewritten for better reader comprehension. An outline or table would help for clearer follow-up.

We appreciate the reviewer’s comments and have made the following revisions in response. We have removed the section of the references that contained supplementary information. Our manuscript now begins directly with Section 1, the Introduction, followed by Sections 2-7. These sections cover: The current HCC surveillance recommendations for CHB patients, The PAGE-B model, The accuracy of the PAGE-B score in predicting HCC risk in CHB patients, Modifications to the PAGE-B score and their predictive utility, Comparisons of PAGE-B with other risk scores for HCC surveillance in CHB patients and Cost evaluation of the PAGE-B score. We have also included a Figure 1 in the Introduction section, summarizing the key points analyzed in the study (page 2).

In Section 6, we provide a detailed report of studies that have investigated the validation and application of the PAGE-B score compared to other risk scores for HCC surveillance in CHB patients. Additionally, we have added Table 3 (page 11), which summarizes the main findings of these studies and Figures 3 and 4 (pages 13-14) which illustrate the accuracy of PAGE-B scores in predicting HCC compared to other HCC risk scores according to the data of the tables.

Reviewer 3 Report

Comments and Suggestions for Authors

Thank you for the opportunity to review this comprehensive manuscript regarding the diagnostic accuracy and utility of the PAGE-B scoring system to predict the risk of HCC in patients with CHB. The manuscript is well designed, presenting accurate and current information regarding the epidemiology of CHB and the associated long-term risk of HCC, with or without treatment or cirrhosis. PAGE-B is defined and introduced within the context of a variety of accepted screening methods, including the advantages of PAGE-B in countries or areas with limited healthcare resources. There is a detailed review of previous studies which analyzed the predictive accuracy of PAGE-B in different countries and patient populations, including those coinfected with HIV.

You state in the second sentence of the Introduction that CHB remains a major cause of morbidity and mortality in Africa and East Asia. While you include studies from Taiwan, Hong Kong, and Japan in your review, there unfortunately appear to be no studies investigating the use of PAGE-B in Africa. Please address the potential impact of this on the potential use of PAGE-B in African patients.

Author Response

Responses to the comments made by Reviewer 3

Thank you for the opportunity to review this comprehensive manuscript regarding the diagnostic accuracy and utility of the PAGE-B scoring system to predict the risk of HCC in patients with CHB. The manuscript is well designed, presenting accurate and current information regarding the epidemiology of CHB and the associated long-term risk of HCC, with or without treatment or cirrhosis. PAGE-B is defined and introduced within the context of a variety of accepted screening methods, including the advantages of PAGE-B in countries or areas with limited healthcare resources. There is a detailed review of previous studies which analyzed the predictive accuracy of PAGE-B in different countries and patient populations, including those coinfected with HIV.

You state in the second sentence of the Introduction that CHB remains a major cause of morbidity and mortality in Africa and East Asia. While you include studies from Taiwan, Hong Kong, and Japan in your review, there unfortunately appear to be no studies investigating the use of PAGE-B in Africa. Please address the potential impact of this on the potential use of PAGE-B in African patients.

We thank the reviewer for these comments. In response, we have added a commentary on the application of the PAGE-B score to patients of African origin in the Conclusions section (pages 14-15).